# A gonad-expressed opsin mediates light-induced spawning in the jellyfish *Clytia*

**Gonzalo Quiroga Artigas[1], Pascal Lapébie[1], Lucas Leclère[1], Noriyo Takeda[2], Ryusaku Deguchi[3], Gáspár Jékely[4,5], Tsuyoshi Momose[1]\*, Evelyn Houliston[1]\***

[1] Laboratoire de Biologie du Développement de Villefranche-sur-mer (LBDV), Sorbonne Universités, UPMC Univ. Paris 06, CNRS, Villefranche-sur-mer, France; [2]Research Center for Marine Biology, Graduate School of Life Sciences, Tohoku University, Aomori, Japan; [3]Department of Biology, Miyagi University of Education, Sendai, Japan; [4]Max Planck Institute for Developmental Biology, Tübingen, Germany; [5]Living Systems Institute, University of Exeter, Exeter, United Kingdom

**Abstract** Across the animal kingdom, environmental light cues are widely involved in regulating gamete release, but the molecular and cellular bases of the photoresponsive mechanisms are poorly understood. In hydrozoan jellyfish, spawning is triggered by dark-light or light-dark transitions acting on the gonad, and is mediated by oocyte maturation-inducing neuropeptide hormones (MIHs) released from the ectoderm. We determined in *Clytia hemisphaerica* that blue-cyan light triggers spawning in isolated gonads. A candidate opsin (Opsin9) was found co-expressed with MIH within specialised ectodermal cells. *Opsin9* knockout jellyfish generated by CRISPR/Cas9 failed to undergo oocyte maturation and spawning, a phenotype reversible by synthetic MIH. Gamete maturation and release in *Clytia* is thus regulated by gonadal photosensory-neurosecretory cells that secrete MIH in response to light via Opsin9. Similar cells in ancestral eumetazoans may have allowed tissue-level photo-regulation of diverse behaviours, a feature elaborated in cnidarians in parallel with expansion of the opsin gene family.
DOI: https://doi.org/10.7554/eLife.29555.001

**\*For correspondence:**
tsuyoshi.momose@obs-vlfr.fr (TM);
houliston@obs-vlfr.fr (EH)

**Competing interests:** The authors declare that no competing interests exist.

## Introduction

Integration of environmental light information contributes to tight coordination of gamete release in a wide range of animal species. The nature of the photodetection systems involved and their evolutionary origins are poorly undestood. Proposed involvement has mainly focussed on light-entrainment of endogenous clocks, which align many aspects of physiology and behaviour, including reproductive ones, to seasonal, monthly or daily cycles (*Cermakian and Sassone-Corsi, 2002*; *Tessmar-Raible et al., 2011*). Clock entrainment can involve both of the main families of photo-sensitive proteins (photopigments) used for animal non-visual photoreception: the evolutionarily ancient Cryptochrome/Photolyase family, which originated in unicellular eukaryotes (*Oliveri et al., 2014*), and the diverse, animal-specific opsin family of light-sensitive G Protein-Coupled Receptors (GPCRs) known best for involvement in visual photodetection (*Cronin and Porter, 2014*; *Gehring, 2014*). Light cues can also provide more immediate triggers for gamete release, which integrate with seasonal and/or circadian regulation (*Kaniewska et al., 2015*; *Lambert and Brandt, 1967*), but the involvement of specific photopigments in such regulation has not previously been addressed.

Members of Hydrozoa, a subgroup of Cnidaria which can have medusae or polyps as the sexual form, commonly display light-regulated sexual reproduction (*Leclère et al., 2016*; *Siebert and Juliano, 2017*). They have simple gonads in which the germ cells are sandwiched between ectoderm and endoderm, and unlike many other animals they lack additional layers of somatic follicle cells surrounding oocytes in the female (*Deguchi et al., 2011*). Light-dark and/or dark-light transitions

**eLife digest** Many animals living in the sea reproduce by releasing sperm and egg cells at the same time into the surrounding water. Animals often use changes in ambient light at dawn and dusk as reliable daily cues to coordinate this spawning behavior between individuals. For example, jellyfish of the species *Clytia hemisphaerica,* which can easily be raised in the laboratory, spawn exactly two hours after the light comes on.

Researchers recently discovered that spawning in *Clytia* and other related jellyfish species is coordinated by a hormone called 'oocyte maturation-inducing hormone', or MIH for short. This hormone is produced by a cell layer that surrounds the immature eggs and sperm within each reproductive organ, and is secreted in response to light cues. It then diffuses both inside and outside of the jellyfish, and triggers the production of mature eggs and sperm, followed by their release into the ocean. However, until now it was not known which cells and molecules are responsible for detecting light to initiate the secretion of MIH.

Quiroga Artigas et al. – including some of the researchers involved in the MIH work – now discovered that a single specialised cell type in the reproductive organs of *Clytia* responds to light and secretes MIH. These cells contain a light-sensitive protein called Opsin9, which is closely related to the opsin proteins in the human eye well known for their role in vision. When Opsin9 was experimentally mutated, *Clytia* cells could not secrete MIH in response to light, and the jellyfish failed to spawn. This opsin protein is thus necessary to detect light in order to trigger spawning in jellyfish.

A next step will be to examine and compare whether other proteins of the opsin family and hormones related to MIH also regulate spawning in other marine animals. This could have practical benefits for raising marine animals in aquariums and as food resources, and in initiatives to protect the environment. More widely, these findings could help unravel how sexual reproduction has evolved within the animal kingdom.

DOI: https://doi.org/10.7554/eLife.29555.002

trigger the release of mature gametes into the seawater by rupture of the gonad ectoderm (*Freeman and Ridgway, 1988*; *Miller, 1979*; *Roosen-Runge, 1962*). Gamete release is coordinated with diel migration behaviours in jellyfish to ensure gamete proximity for fertilisation (*Martin, 2002*; *Mills, 1983*). We know that the photodetection systems that mediate hydrozoan spawning operate locally within the gonads, since isolated gonads will spawn upon dark-light or light-dark transitions (*Freeman, 1987*; *Ikegami et al., 1978*). Opsin gene families have been identified in cnidarians and provide good candidates for a role in this process, with expression of certain opsin genes reported in the gonads both of the hydrozoan jellyfish *Cladonema radiatum* (*Suga et al., 2008*) and of the cubozoan jellyfish *Tripedalia cystophora* (*Liegertová et al., 2015*).

In female hydrozoans, the regulation of spawning is tightly coupled to oocyte maturation, the process by which resting ovarian oocytes resume meiosis to be transformed into fertilisable eggs (*Yamashita et al., 2000*). Oocyte maturation, followed by spawning, is induced by 'Maturation Inducing Hormones' (MIHs), released from gonad somatic tissues upon reception of the appropriate light cue (*Freeman, 1987*; *Ikegami et al., 1978*). The molecular identity of MIH of several hydrozoan species has been uncovered recently as PRPamide family tetrapeptides, which are released from scattered cells with neural-type morphology present in the gonad ectoderm following dark-light transitions (*Takeda et al., 2018*). Here we report the identification of a cnidarian opsin gene expressed in the gonads of the hydrozoan jellyfish *Clytia hemisphaerica* and show by gene knockout that it is essential for spawning via MIH release in response to light. We further show that this opsin is expressed in the same gonad ectoderm cells that secrete MIH, which thus constitute a specialised cnidarian cell type responsible for mediating light-induced spawning. We discuss how these cells show many features in common with photosensitive deep brain neuroendocrine cells, such as those described in fish and annelids (*Tessmar-Raible et al., 2007*). These cell types may have shared an origin in the common ancestor of the cnidarians and of the bilaterians, an animal clade comprising all the protostomes and deuterostomes.

## Results

### Spawning *of Clytia* ovaries is induced by blue-cyan light

MIH release in *Clytia* gonads is triggered by a light cue after a minimum dark period of 1–2 hr, with mature eggs being released two hours later (*Amiel et al., 2010*). In order to characterise the light response of *Clytia* gonads (*Figure 1A*), we first assessed the spectral sensitivity of spawning. Groups of 3–6 manually dissected *Clytia* female gonads were cultured overnight in the dark and then stimulated from above with 10 s pulses of monochromatic light across the 340 to 660 nm spectrum. Stimulated ovaries were returned to darkness for one hour before scoring oocyte maturation. Oocyte maturation is accompanied by breakdown of the membrane of the oocyte nucleus ('GV' for germinal vesicle) in fully-grown oocytes, followed by polar body emmission and spawning. We found that wavelengths between 430 and 520 nm provoked spawning in at least 50% of gonads, with 470–490 nm wavelengths inducing spawning of ≥75% of gonads (*Figure 1B*). Oocyte maturation and subsequent spawning of *Clytia* female gonads is thus preferentially triggered by blue-cyan light (*Figure 1B*), the wavelength range which penetrates seawater the deepest (*Gehring and Rosbash, 2003*; *Gühmann et al., 2015*).

### A highly expressed opsin in *Clytia* gonad ectoderm

We identified a total of ten *Clytia* opsin sequences in transcriptome and genome assemblies from *Clytia* by reciprocal BLAST searches using known hydrozoan opsin sequences (*Suga et al., 2008*) and by the presence of the diagnostic lysine in the seventh transmembrane domain to which the retinal chromophore binds (*Terakita et al., 2012*) (*Figure 2—figure supplement 1*). We selected candidate opsins for a role in mediating MIH release by evaluating expression in the isolated gonad ectoderm, a tissue which has been shown using *Cladonema* jellyfish to have an autonomous capacity to respond to light (*Takeda et al., 2018*). Illumina HiSeq reads from *Clytia* gonad ectoderm, gonad endoderm, growing and fully grown oocyte transcriptome sequencing were mapped against each opsin sequence (*Figure 2A*). In the gonad ectoderm sample, two opsin mRNAs (Opsin4 and Opsin7) were detected at low levels and one was very highly expressed (Opsin9), while in the three other gonad samples analysed opsin expression was virtually undetectable (*Figure 2A*). The high expression of *Opsin9* gene in the gonad ectoderm made it a strong candidate for involvement in light-induced spawning.

### *Opsin9* is expressed in MIH-secreting gonad ectoderm cells

In situ hybridisation of *Clytia* ovaries revealed *Opsin9* expression in a scattered population of gonad-ectoderm cells (*Figure 2B*). No *Opsin9* expression was detected anywhere else in the medusa. We were unable to detect by in situ hybridisation in *Clytia* gonads either of the two lowly-expressed gonad-ectoderm opsin genes, *Opsin4* and *Opsin7* (*Figure 2D–E*). The distribution of *Opsin9*-expressing cells was highly reminiscent of the expression pattern for *PP1* and *PP4* (*Figure 2C*), the two MIH neuropeptide precursor genes co-expressed in a common gonad ectoderm cell population (*Takeda et al., 2018*). Double fluorescent in situ hybridisation using probes for *Opsin9* and for *PP4* revealed that these genes were expressed in the same cells (*Figure 2F*); 99% of Opsin9 mRNA positive cells were also positive for PP4 mRNA, and over 87% of *PP4 mRNA*-positive cells were also positive for *Opsin9* mRNA. This finding raised the possibility that spawning in *Clytia* might be directly controlled by light detection through an opsin photopigment in the MIH-secreting cells of the gonad ectoderm.

### *Opsin9* gene knockout prevents oocyte maturation and spawning

To test the function of Opsin9 in light-induced oocyte maturation and spawning, we generated a *Clytia Opsin9* knockout (KO) polyp colony using a CRISPR/Cas9 approach, which produces very extensive bi-allelic KO of the targeted gene in medusae of the F0 generation (*Momose and Concordet, 2016*; see Materials and methods for details). This approach is favoured by the *Clytia* life cycle, in which larvae developed from each CRISPR-injected egg metamorphose into a vegetatively-expanding polyp colony, from which sexual medusae bud clonally (*Houliston et al., 2010*; *Leclère et al., 2016*). CRISPR guide RNAs were designed to target a site in the first exon of *Opsin9* encoding the third transmembrane domain, and were verified not to match any other sites in

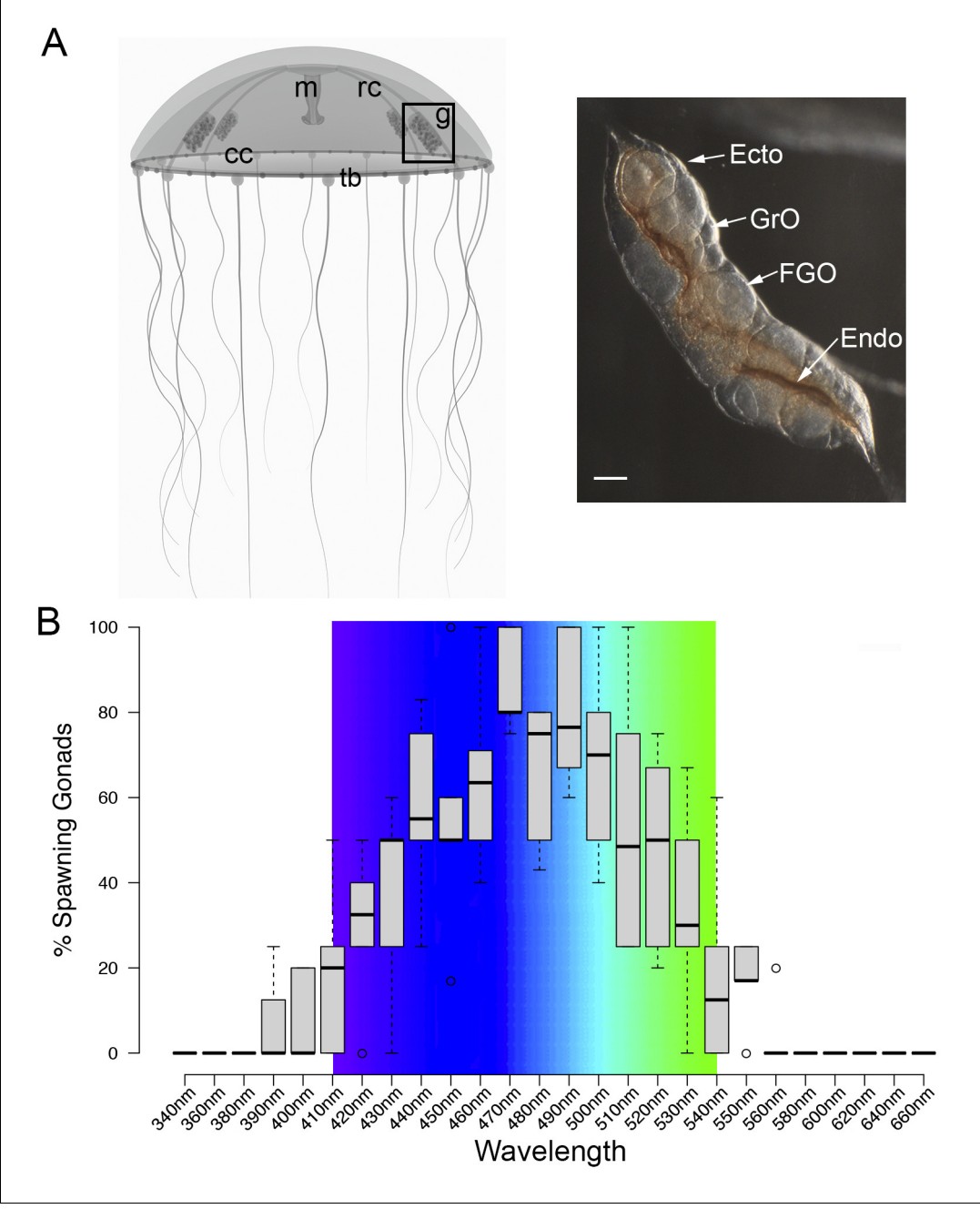

**Figure 1.** Spectral characterisation of spawning in *Clytia* ovaries. (**A**) Schematic of a *Clytia hemisphaerica* female jellyfish: The four gonads (g) lie on the radial canals (rc) that connect the manubrium (M) to the circular canal (cc) running around the bell rim, adjacent to the tentacle bulbs (tb). (Inset) Photo of a *Clytia* ovary. FGO = fully grown oocytes; GrO = growing oocytes; Endo = gonad endoderm; Ecto = gonad ectoderm. Bar = 100 μm. (**B**) Box Plot showing spectral characterisation of *Clytia* spawning. Groups of 3–6 isolated gonads were exposed to 10 s pulses of monochromatic light (see Materials and methods). Gonads were considered to spawn if at least one oocyte underwent maturation and release. Statistics were based on percentage of gonad spawning in response to a specific wavelength obtained from 5 to 6 independent experiments. A total of 20–30 gonads were analysed per wavelength. Centre lines show the medians; box limits indicate the 25th and 75th percentiles (first and third quartiles); whiskers extend 1.5 times the interquartile range from the 25th and 75th percentiles; outliers are represented by circles. Colour spectrum is shown for 410 nm – 540 nm wavelengths.
DOI: https://doi.org/10.7554/eLife.29555.003

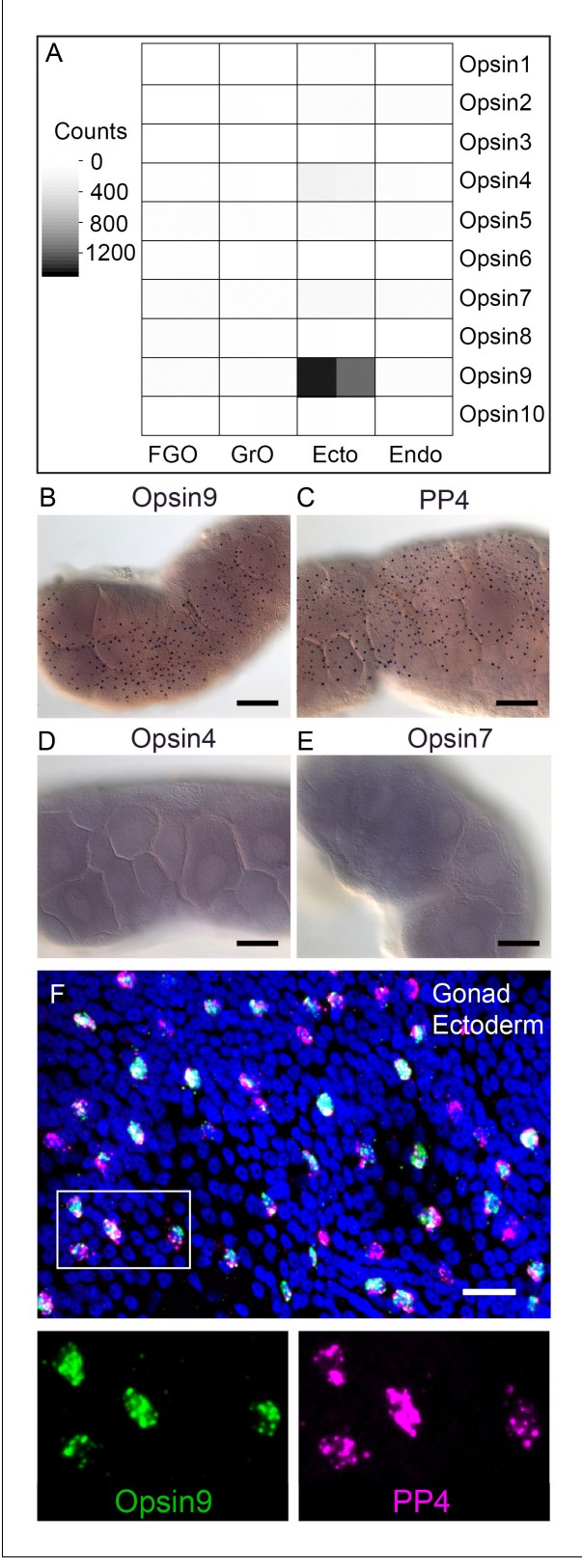

**Figure 2.** *Clytia Opsin* expression in gonad ectoderm cells. (**A**) Heat map representing the expression of the ten opsin sequences from *Clytia hemisphaerica* in different gonad tissues. Illumina HiSeq 50nt reads from isolated endoderm (Endo), ectoderm (Ecto), growing (GrO) and fully-grown oocytes (FGO) from mature female ovaries were mapped against the opsin sequences. Counts were normalised per total number of reads in each sample

*Figure 2 continued on next page*

*Figure 2 continued*

and per sequence length (*Figure 2—source data 1*). (**B**) In situ hybridisation (ISH) detection of *Opsin9* mRNA in scattered ectodermal cells of female *Clytia* gonads. (**C**) ISH of the neuropeptide precursor *PP4* mRNA in *Clytia* female gonads, also showing a scattered pattern in the ectoderm. (**D-E**) ISH of *Opsin4* and *Opsin7* mRNAs, respectively, in *Clytia* ovaries, showing no detectable localised expression of these two opsin genes. (**F**) Double fluorescent ISH showing co-expression of *Opsin9* (green) and *PP4* (magenta) mRNAs in gonad ectoderm cells; nuclei (Hoechst) in blue. Single channels are shown for the outlined zone in the top image. Of n = 594 randomly chosen cells expressing either gene counted in 10 different gonads, over 86% co-expressed Opsin9 and PP4 mRNAs. Controls with single probes were performed to validate a correct fluorescence inactivation and ensure that the two channels did not cross-over (not shown). Scale bars in B-E = 100 µm; F = 20 µm.

DOI: https://doi.org/10.7554/eLife.29555.004

The following source data and figure supplement are available for figure 2:

**Source data 1.** Expression data used to construct the heatmap in *Figure 2A*.

DOI: https://doi.org/10.7554/eLife.29555.006

**Figure supplement 1.** Alignment of selected opsin sequences, highlighting amino acids crucial for light detection and opsin function.

DOI: https://doi.org/10.7554/eLife.29555.005

---

available *Clytia* genome sequences. One of the polyp colonies generated carried a predominant 5 bp deletion, corresponding to a frame-shift and premature STOP codon (*Figure 3A*). Polyp colony development in this individual, designated *Opsin9*$^{n1-4}$ (see Materials and methods), showed no abnormal features.

For phenotypic analysis we collected *Opsin9*$^{n1-4}$ jellyfish, which were all females, and grew them by twice-daily feeding for two weeks to sexual maturity. Although these *Opsin9*$^{n1-4}$ mature medusae initially appeared normal, they did not spawn after the daily dark-light transition, and after a few days displayed grossly inflated ovaries due to an accumulation of unreleased large immature oocytes with intact GVs (*Figure 3B–C*). In three independent experiments to test the light response of isolated gonads, over 85% of *Opsin9*$^{n1-4}$ gonads failed to undergo oocyte maturation and spawning upon light stimulation (*Figure 3D*). *Opsin9*$^{n1-4}$ gonads did, however, release oocytes in response to synthetic MIH peptide (*Figure 3E*). These oocytes resumed meiosis normally and could be fertilised, although the fertilisation rate was lower than for oocytes spawned in parallel from wild type gonads, possibly a consequence of their prolonged retention in the gonad.

Genotyping of individual gonads showed that the rare gonads from *Opsin9*$^{n1-4}$ medusae that spawned after light had greater mosaicism of mutations, with a higher ratio of residual non-frame-shift mutations and also a significant amount of wild type cells, whereas gonads that failed to spawn carried mainly the predominant 5 bp deletion, a second 21 bp deletion and no detectable wild type cells (*Figure 3—figure supplement 1*).

The failure of spawning observed in *Clytia Opsin9*$^{n1-4}$ mutant jellyfish gonads, together with the absence of detectable *Opsin9* expression in non-gonad tissues of the medusa and the autonomous response of isolated wild-type gonads to light, strongly indicates that the gonad photopigment Opsin9 plays an essential role in light-induced oocyte maturation.

## Opsin9 is required for light-induced MIH secretion from gonad ectoderm

Since Opsin9 and MIH are co-expressed in the same cells, we reasoned that *Opsin9* function was probably required for MIH secretion. This was confirmed by immunofluorescence of *Opsin9* mutant gonads (*Figure 4*). Quantitative immunofluorescence analyses based on anti-MIH staining were performed in both wild type and *Opsin9* mutant gonads, comparing light-adapted and dark-adapted specimens fixed 45 min after white light exposure. MIH-Opsin9 cells were identified by their characteristic organisation of stable microtubules (*Takeda et al., 2018*; see Figure 5E-F). Whereas wild type gonads exhibited a significant decrease of MIH fluorescence values in MIH-Opsin9 cells upon light stimulation (*Figure 4A–C*), *Opsin9* mutant gonads maintained similar levels of MIH fluorescence in both conditions (*Figure 4D–F*). Moreover, average MIH fluorescence levels per cell were significantly higher (U test at p<0.01) in *Opsin9* mutant than in wild type ovaries suggesting a progressive accumulation of MIH in *Opsin9* mutant gonads.

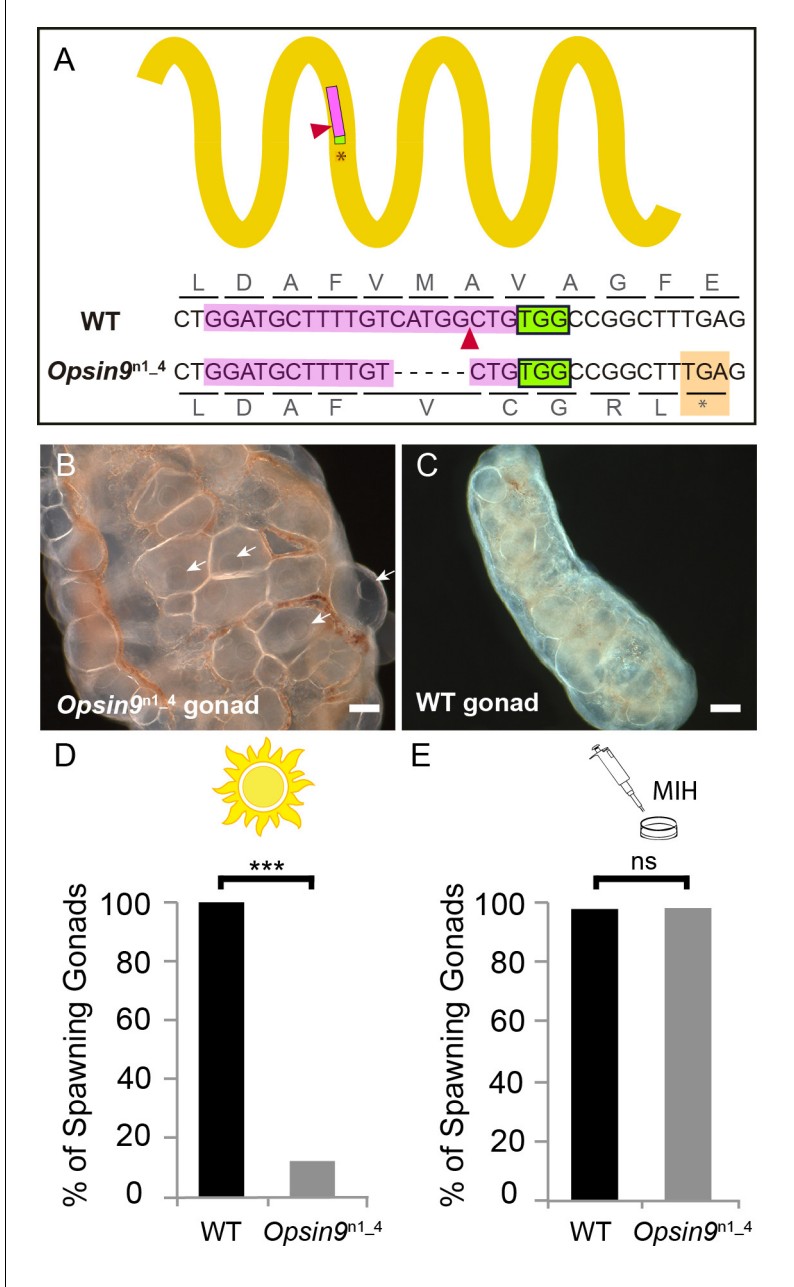

**Figure 3.** Production and phenotype of *Opsin9* knockout medusae. (**A**) Scheme of Opsin9 GPCR showing part of the genomic region coding for the third transmembrane domain targeted by *Opsin9* n1 CRISPR sgRNA. Corresponding amino acids are shown. Pink boxes indicate the target site of the sgRNA. Green boxes are the PAM sequence (NGG). The expected cleavage site of Cas9 is indicated by red triangles. The predominant 5 bp deletion detected in the *Opsin9*$^{n1-4}$ mutant is shown. This mutation leads to a frame-shift and an early STOP codon in *Opsin9*$^{n1-4}$ (orange box). (**B**) Highly inflated gonad of an *Opsin9*$^{n1-4}$ mutant female jellyfish showing the abnormal accumulation of large oocytes with intact GVs (arrows). (**C**) Wild type *Clytia* gonad at the same magnification. Images in B and C were both taken 8 hr after the natural light cue, accounting for the absence of fully-grown oocytes in the wild type gonad. Scale bars all 100 μm. (**D**) Quantification of spawning upon light stimulation of wild type (WT) and *Opsin9*$^{n1-4}$ gonads. Percentage of spawning gonads combined from three independent experiments is shown in all cases; n = 92 for wild type and n = 154 gonads for *Opsin9*$^{n1-4}$ mutants. (**E**) Equivalent analysis for synthetic MIH treatment of wild type and *Opsin9*$^{n1-4}$ gonads. Oocyte maturation and spawning were induced by synthetic MIH treatment in both cases; n = 94 gonads for wild type and n = 80 gonads for *Opsin9*$^{n1-4}$ mutants. Fisher's exact test showed a significant difference (at p<0.01) between spawning in wild type and mutant samples stimulated by light (**D**), but not by MIH (**E**).

*Figure 3 continued on next page*

*Figure 3 continued*

DOI: https://doi.org/10.7554/eLife.29555.007

The following source data and figure supplement are available for figure 3:

**Source data 1.** Sequences of guide RNAs and genotyping primers used in this study.

DOI: https://doi.org/10.7554/eLife.29555.009

**Figure supplement 1.** $Opsin9^{n1-4}$ genotyping results.

DOI: https://doi.org/10.7554/eLife.29555.008

These immunofluorescence results indicated that *Opsin9* mutant medusae failed to undergo oocyte maturation and spawning because of reduced MIH secretion in response to light. The ability of synthetic MIH peptides to reverse the phenotype of *Opsin9* mutant gonads supports this conclusion: Equivalent MIH concentrations to those effective for light-adapted isolated gonads (*Takeda et al., 2018*) reliably induced oocyte maturation and spawning in *Opsin9* mutant isolated gonads (*Figure 3E*). The primary defect in these *Opsin9* mutants is thus in MIH secretion from the ectoderm following light stimulation, providing strong evidence that the CRISPR-Cas9 approach had specifically targeted the Opsin9 gene.

Taken together, these results demonstrate that Opsin9 is required for the MIH-producing cells in the *Clytia* gonad to release the MIH neuropeptides following dark-light transitions, and thus has an essential role in light-dependent reproductive control.

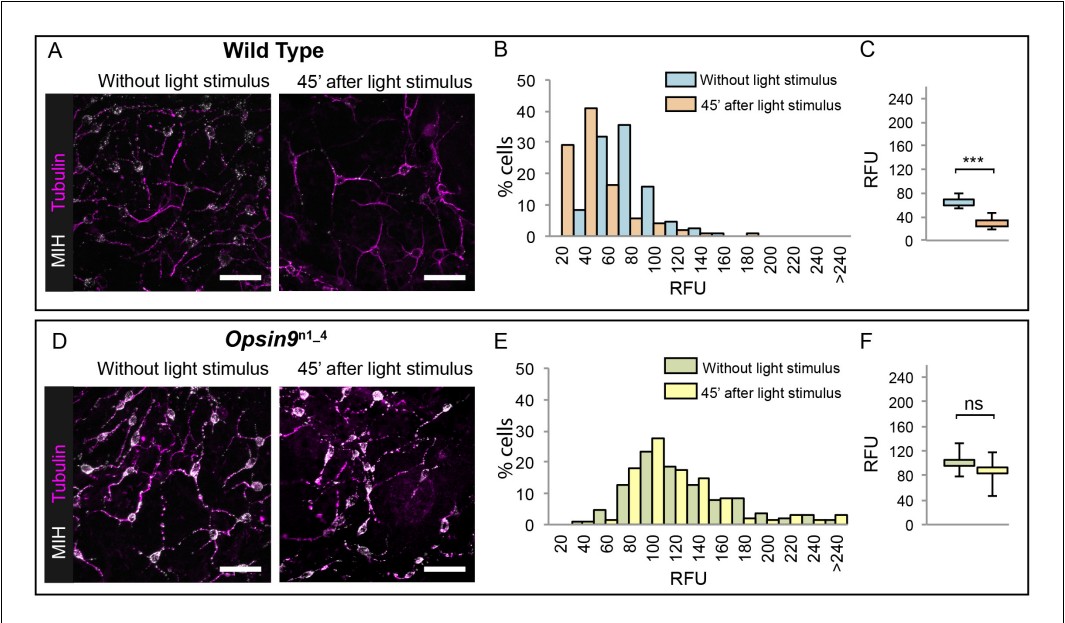

**Figure 4.** Accumulation of MIH in *Opsin9* mutant gonads. Quantification of MIH loss from gonad ectoderm after light stimulation. All images were summed from 10 confocal Z planes over 4 μm, acquired on the same day using constant settings. Immunofluorescence of fixed, methanol –treated gonads using anti-PRPamide antibodies (MIH; white) was quantified in putative MIH-Opsin cells identified by typical morphology revealed by anti-alpha tubulin (magenta) antibodies. (**A**) Representative images of wild type *Clytia* gonad showing reduction in MIH fluorescence after light stimulation. (**B**) Distribution of RFU (Relative Fluorescence Units) values obtained for each cell analysed in the two conditions (number of cells analysed: n = 226 and n = 282, respectively). (**C**) Graph showing the medians of the data in (**B**). Limits correspond to first and third quartiles. The Mann-Whitney U test showed a significant difference between conditions (at p<0.01). (**D**) Representative immunofluorescence images of $Opsin9^{n1-4}$*Clytia* gonad MIH-secreting cells before and after light stimulation. MIH fluorescence is maintained upon light stimulation. (**E**) Distribution of RFU values after fluorescence quantification in $Opsin9^{n1-4}$ putative MIH-Opsin cells in the two conditions (n = 183 and n = 201, respectively). (**F**) Graph showing the medians and quartiles of the data in (**E**). Mann-Whitney U test did not show a significant difference at p<0.01. Scale bars all 20 μm.

DOI: https://doi.org/10.7554/eLife.29555.010

## Neural type morphology of *Clytia* gonad MIH-secreting cells

The functional studies described above indicate that *Clytia* gonad cells that co-express MIH and Opsin9 have photosensory functions, as well as neurosecretory characteristics (*Takeda et al., 2018*). To investigate the morphology of these key cells in more detail we performed immunofluorescence to visualise cortical actin and microtubules in cells producing MIH (*Figure 5*). The *Clytia* gonad ectoderm consists of a monolayer of ciliated epitheliomuscular cells (*Leclère and Röttinger, 2017*), tightly joined by apical junctions and bearing basal extensions containing muscle fibres. These basal myofibres form a layer over the oocytes and stain strongly with phalloidin fluorescent probes (*Figure 5A–D*). The cell bodies of the MIH-secreting cells were found to be positioned within this epithelial layer, scattered between the much more abundant epitheliomuscular cells (*Takeda et al., 2018*; *Figure 5A,B,D*). A variable number of neural-type processes containing prominent microtubule bundles project basally from these cells (*Figure 5E,F*), and intermingle with the muscle fibres of surrounding epitheliomuscular cells (*Figure 5A–C*). The cell nuclei were generally located more basally than those of the surrounding epitheliomuscular cells (*Figure 5A,C*). In regions overlying large oocytes, the gonad ectoderm, including most of the MIH-rich basal processes, was elevated by a characteristic space on the basal side (*Figure 5C*), probably reflecting the accumulation of extracellular jelly components around oocytes during late stages of growth. This configuration implies that most of the peptide hormone (MIH) is secreted at distance from its site of action at the oocyte surface, consistent with a neuroendocrine function. MIH-positive cells did not have extensive apical domains exposed on the external face of the ectodermal epithelium, although MIH staining was in some cases detected very close to this surface at the interstices between surrounding myoepithelial cells (*Figure 5C' and D"*). Cilia, which decorated most epitheliomuscular cells, could not be unambiguously associated with the MIH-secreting cells (*Figure 5F,G*), but basal bodies could be detected in the apical side of these cells with a gamma-tubulin antibody (*Figure 5H–H'*).

These various immunofluorescence analyses indicate that the MIH-secreting cells have morphological features characteristic of multipolar neurosensory cells in cnidarians (*Saripalli and Westfall, 1996*). Other approaches such as electron microscopy will be required to resolve important questions concerning these cells, notably whether closely apposed processes from adjacent cells (*Figure 5E,F*) connect via synapses to form a functional network, and whether they are fully integrated into the epithelial ectoderm through stable junctions and distinct apical surfaces. Nevertheless, based on opsin and neuropeptide expression, general morphology, and biological function, we can confidently propose that this specialised cell type has a dual sensory-neurosecretory nature.

## Evolution of the *Clytia Opsin9* gene

With increasing availability of opsin gene sequences covering a widening range of animal taxa, the traditional division into 'c-opsins', expressed in ciliary photoreceptor cells such as vertebrate rods and cones, and 'r-opsins,' as expressed in arthropod rhabdomeric photoreceptors, is clearly no longer adequate as a representation of opsin phylogenetic relationships (*Cronin and Porter, 2014*; *Feuda et al., 2012*, *2014*). Molecular phylogeny analyses are progressively revealing a complicated evolutionary history for the opsins, involving many gene duplications and losses in separate animal lineages, starting from a set of at least nine genes in the common ancestor of Bilateria, with three or four sequences already present before the cnidarian-bilaterian split (*Ramirez et al., 2016*; *Vöcking et al., 2017*). Available cnidarian opsin sequences form three groups, termed 'anthozoan opsin I', 'anthozoan opsin II' and 'cnidops'. The first two contain only anthozoan (coral and sea anemone) sequences. The cnidops group includes some anthozoan opsins and all published opsin genes from the medusozoan clade (jellyfish and hydra), and is probably the sister group to the xenopsin group, which includes opsins from a few disparate protostome taxa (*Ramirez et al., 2016*; *Vöcking et al., 2017*).

We performed molecular phylogeny for the *Clytia* opsin genes by adding the ten sequences to two recent datasets of animal opsin amino acid sequences (*Feuda et al., 2014*; *Vöcking et al., 2017*) and using the LG+Γ amino acid substitution model. All 10 *Clytia* Opsins fell as expected within the cnidops group, with Opsin9 and the closely-related Opsin10 having more divergent sequences than the other *Clytia* opsins (*Figure 6A*, *Figure 6—figure supplement 1*). Some trials, for instance using the GTR+Γ model (conceived for large datasets) on the *Vöcking et al. (2017)* based alignment, produced an alternative tree topology with *Clytia* Opsin9/10 positioned as sister-group

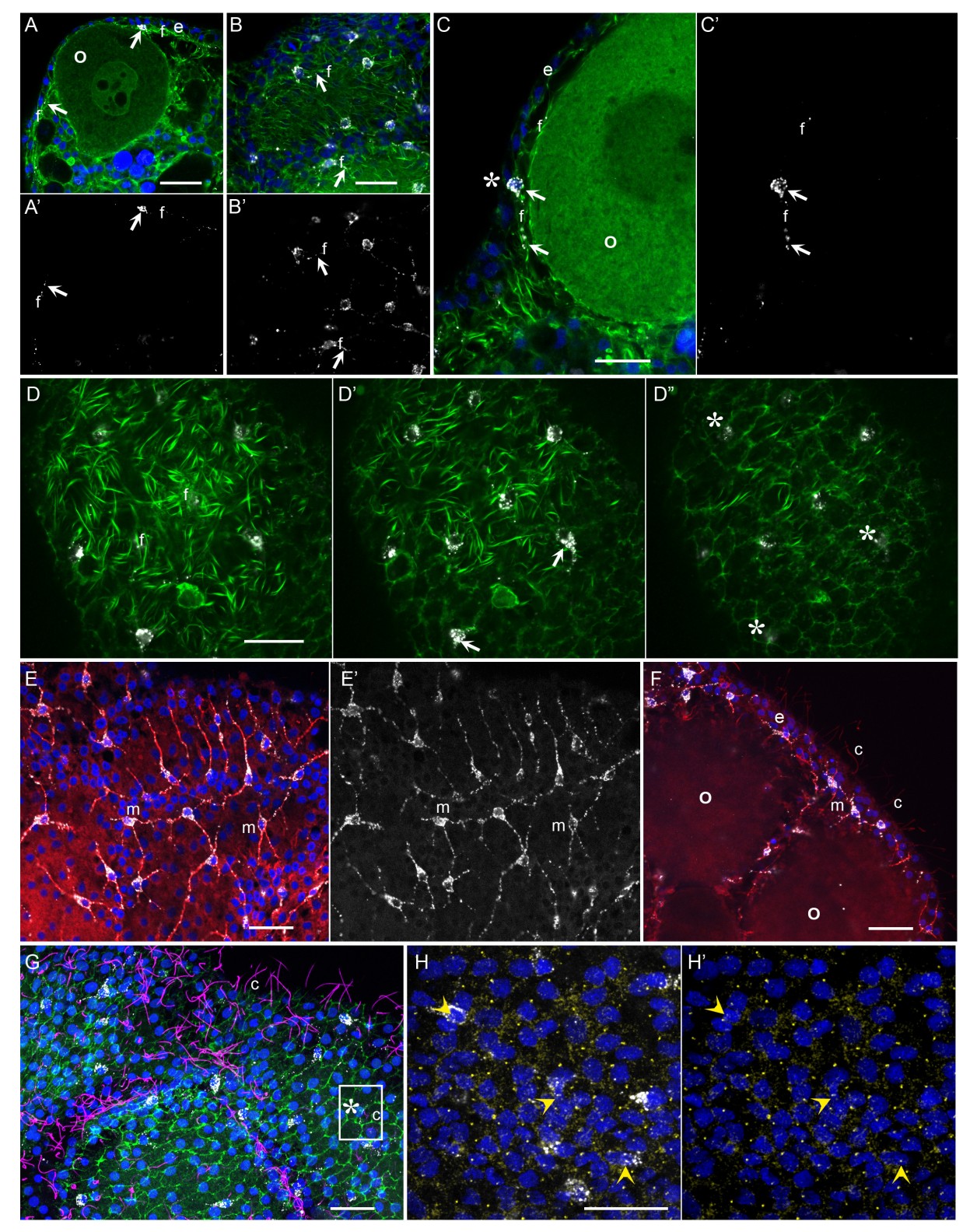

**Figure 5.** Morphology of *Clytia* MIH-secreting cells. Confocal images of fluorescence stained isolated *Clytia* ovaries. Throughout the figure, anti-PRPamide (MIH) staining is shown in white and Hoechst staining of DNA in blue. Colours and annotations are consistent across panels. A', B' and E' show PRPamide staining only , corresponding to the overlays in A, B and E. (A-D) Relationship between MIH-producing cells and the myoepithelial cells of the gonad ectoderm revealed by phalloidin staining of f-actin (green). The position of the basal layer of myofibrils is indicated by f, and examples of

*Figure 5 continued on next page*

*Figure 5 continued*

anti-PRPamide stained cell bodies and basal extensions are shown by white arrows. (**A**) Cross-section through an early growth-stage oocyte (o) and overlying ectoderm (e). (**B**) Glancing section through the ectodermal layer covering a largeroocyte, illustrating the characteristic phalloidin staining of basal myofibrils (f) in the centre of the image, and of the polygonal network of apical junctions in the peripheral parts of the image. (**C**) Higher magnification cross section of a late growth-stage oocyte and overlying ectoderm. The apical tip of anti-PRPamide stained cell lies close to the exposed surface of the ectoderm (asterisk), and the basal projections run in the myofibril layer. (**D-D'**) Three confocal sections taken at 1 μm intervals at progressively more superficial levels of gonad ectoderm, positioned above a large oocyte. The basal myofibril layer of the epitheliomuscular cells predominates in D and their apical junctions in D'. MIH cell bodies are centred between these two layers (**D'**). At the gonad surface (**D"**) anti-PRPamide staining is detectable at interstices between ectodermal cells (asterisks). (**E-F**) Anti-alpha tubulin staining (red) of methanol extracted samples highlights the microtubule bundles characteristic of the basal processes of the multipolar PRPamide-stained cells (m) and apical cilia (c) of the ectodermal cells. (**E**) Confocal plane at the basal ectoderm level. (**F**) maximum projection of 5 confocal planes (0.8 μm intervals) through oocytes and overlying ectoderm. (**G**) Anti- acetylated alpha-tubulin (magenta) of cilia combined with phalloidin staining (green) in a grazing superficial confocal section. A possible example of a cilium associated with the apical tip of a PRPamide-stained cell is highlighted. (**H**) Anti-γ tubulin staining (yellow) in a superficial confocal section showing apical ciliary basal bodies in each epitheliomuscular cell and in PRPamide-stained cells (arrowheads) - image overlaid in H but not H'. Scale bars all 20 μm.

DOI: https://doi.org/10.7554/eLife.29555.011

to the anthozoa opsin II group (*Figure 6—figure supplement 2*), but we could show that this is a 'long branch attraction' artifact. Thus, this topology could be reversed by breaking the long *Clytia* Opsin9/10 branch using a single Opsin9/10 related sequence from unpublished transcriptome data of the hydrozoan medusa *Melicertum octocostatum* (*Figure 6—figure supplement 2*). More detailed analyses including only medusozoan (hydrozoan and cubozoan) opsin sequences available in GenBank (*Figure 6B*) confirmed that *Clytia* Opsin9 and Opsin10 were indeed amongst the most divergent Cnidops sequences. The draft *Clytia* genome sequence (*Leclère et al., 2016*) further revealed that *Clytia* Opsin9 and *Opsin10* genes contain a distinct intron, unlike all other described medusozoan Cnidops genes, in a distinct position to those found in available Anthozoan Opsin II, Anthozoan Cnidops and Xenopsin genes (*Vöcking et al., 2017*; *Liegertová et al., 2015*).

Despite this picture of rapid evolutionary divergence of *Clytia* Opsin9/Opsin10 within the diversifying cnidops family, the *Clytia* Opsin9 amino acid sequence exhibits all the hallmarks of a functional photopigment (*Figure 2—figure supplement 1*). It has conserved amino acids at positions required for critical disulphide bond formation and for Schiff base linkage to the retinal chromophore (*Fischer et al., 2013*; *Gehring, 2014*; *Schnitzler et al., 2012*), including acidic residues at the both potential 'counterion' positions, only one of which is largely conserved in cnidarian opsin sequences (*Liegertová et al., 2015*). The Glu/Asp-Arg-Tyr/Phe motif adjacent to the third transmembrane domain, involved in cytoplasmic signal transduction via G proteins (*Fischer et al., 2013*; *Kojima et al., 2000*; *Schnitzler et al., 2012*) is also present in Opsin9.

The exact relationship of *Clytia* Opsin9/10 to other medusozoan opsins cannot be resolved with the available data. Their phylogenetic position was found to be unstable when analysed using different evolutionary models and datasets (*Figure 6*, *Figure 6—figure supplement 1*, *Figure 6—figure supplement 2*). Extensive sampling from more medusozoan species will be needed to fully resolve their phylogenetic history. We could show unambiguously, however, from our phylogenetic reconstructions (*Figure 6B*) and AU ('approximately unbiased') phylogenetic tests (*Shimodaira, 2002*), that *Clytia* Opsin9 is not orthologous to the opsin genes previously identified as expressed in the gonad of the hydromedusa *Cladonema* (AU test: $p<1e^{-7}$) or the cubomedusa *Tripedalia* (AU test: $p<1e^{-20}$). Thus diversification of the cnidops gene family in Medusozoa was accompanied by expression of different opsin paralogs in the gonad in different species.

To summarise the results of this study, we have provided the first demonstration, using CRISPR/Cas9–mediated gene knockout, of an essential role for an opsin gene in non-visual photodetection in a cnidarian. *Clytia* Opsin9 is required for a direct light-response mechanism that acts locally in the gonad to trigger gamete maturation and release (*Figure 7*). It occurs in specialised sensory-secretory cells of the *Clytia* gonad ectoderm to trigger MIH secretion from these cells upon light reception. This peptidic MIH in turn induces oocyte maturation in females, and also release of motile sperm in males (*Takeda et al., 2018*), efficiently synchronising spawning to maximise reproductive success.

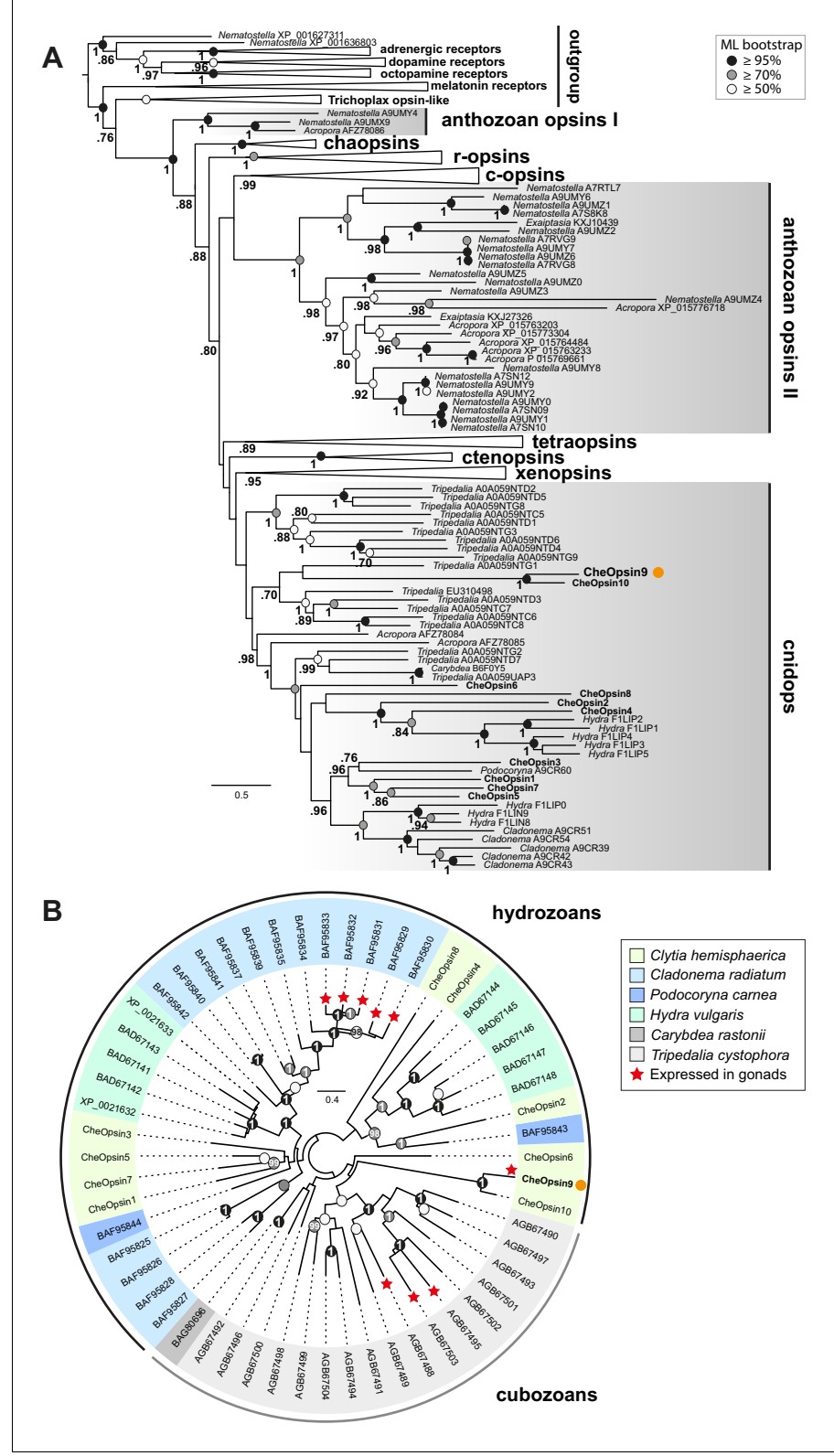

**Figure 6.** Phylogenetic position of *Clytia* opsins. (**A**) Maximum likelihood (ML) phylogenetic analysis (RAxML, LG +Γ, *Figure 6—source data 2*) of opsin proteins based on the dataset of *Vöcking et al. (2017)* including the 10 *Clytia* opsins, rooted with GPCR families closely related to the opsin family. (**B**) Unrooted ML phylogenetic analysis of available medusozoan opsins (RAxML, LG+Γ, *Figure 6—source data 3*). In both trees, ML bootstrap support

*Figure 6 continued*

values (500 replicates) are shown as circles on the branch tips: black circles ≥95%, grey circles ≥70%, white circle ≥50%. Bayesian analyses (MrBayes, LG+Γ) reconstructed trees with similar topologies; Bayesian posterior probabilities greater than 0.70 (**A**) or 0.95 (**B**) are shown next to the branches (**A**) or on the nodes (**B**). The orange dot highlights CheOpsin9. NCBI or Uniprot reference numbers and *Clytia* opsin names are shown. In (**A**), c-opsins = ciliary opsins; tetraopsins = retinal G protein-coupled receptor, Neuropsin and Go-opsin as defined by *Ramirez et al., 2016*; r-opsins = rhabdomeric opsins; ctenopsins = ctenophore opsins; chaopsin = as defined by *Ramirez et al., 2016*. In (**B**), a red star indicates expression in the gonad. Scale bars: estimated number of substitution per site.

DOI: https://doi.org/10.7554/eLife.29555.012

The following source data and figure supplements are available for figure 6:

**Source data 1.** Sequence alignment used for the phylogenetic analyses shown in *Figure 6—figure supplement 2B*, The ten *Clytia* opsin sequences, as well as an Opsin9/10 related sequence from unpublished transcriptome data of the hydrozoan medusa *Melicertum octocostatum*, were added to a published dataset (*Vocking et al., 2017*).

DOI: https://doi.org/10.7554/eLife.29555.015

**Source data 2.** Sequence alignment used for the phylogenetic analyses shown in *Figure 6A* and its *Figure 6—figure supplement 2A*, The ten *Clytia* opsin sequences were added to a published dataset (*Vocking et al., 2017*).

DOI: https://doi.org/10.7554/eLife.29555.016

**Source data 3.** Sequence alignment used for the phylogenetic analyses shown in *Figure 6B*, including the ten *Clytia* opsin sequences and medusozoan opsins available in GeneBank.

DOI: https://doi.org/10.7554/eLife.29555.017

**Source data 4.** Sequence alignment used for the phylogenetic analyses shown in *Figure 6—figure supplement 1A and B*, The ten *Clytia* opsin sequences were added to a published dataset (*Feuda et al., 2014*).

DOI: https://doi.org/10.7554/eLife.29555.018

**Figure supplement 1.** Phylogenetic analyses of the *Clytia* opsins based on the *Feuda et al., (2014)* dataset.

DOI: https://doi.org/10.7554/eLife.29555.013

**Figure supplement 2.** Use of the GTG+Γ amino acid substitution model on the 'source data 1' alignment generates a Long Branch Attraction artefact.

DOI: https://doi.org/10.7554/eLife.29555.014

# Discussion

Comparison of neuropeptide involvement in hydrozoan, starfish, fish and frog reproduction suggested an evolutionary scenario in which gamete maturation and release in ancient metazoans was triggered by gonad neurosecretory cells (*Takeda et al., 2018*). According to this scenario, this cell type would have been largely conserved during cnidarian evolution and function similarly today in hydrozoans, whereas during bilaterian evolution further levels of regulation were inserted between the secreting neurons and the responding gametes. Thus in vertebrates, peptide hormone secretion would have been delocalised to the hypothalamus, and the primary responding cells to the pituitary. Our study on opsins has further revealed a parallel between the MIH-secreting cells of the gonad ectoderm in *Clytia* and deep brain photoreceptor cells in vertebrates, as well as equivalent cells in various protostome species that regulate physiological responses through neurohormone release in response to ambient light (*Fernandes et al., 2013*; *Fischer et al., 2013*; *Halford et al., 2009*; *Tessmar-Raible et al.,*

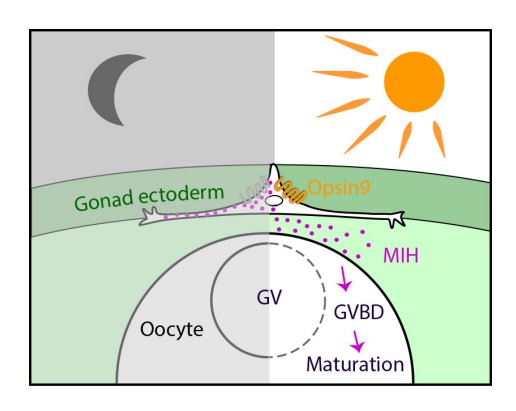

**Figure 7.** Model for Opsin9 function in *Clytia* oocyte maturation. Model for *Clytia* oocyte maturation. At dawn, light activates Opsin9 in specialised cells of the gonad epithelium, causing secretion of MIH inside the gonad. MIH in turn acts on the oocyte surface to trigger the resumption of meiosis, followed by Germinal Vesicle breakdown (GVBD) and oocyte maturation.

DOI: https://doi.org/10.7554/eLife.29555.019

*2007*). Like the *Clytia* MIH-producing cells, TSH-producing cells in birds and fish pituitary (*Nakane et al., 2013*; *Vigh et al., 2002*), and vasopressin/oxytocin-expressing cells in fish hypothalamus and annelid forebrain (*Tessmar-Raible et al., 2007*), show opsin-related photodetection, secrete neuropeptide hormones and are implicated in hormonal control of reproduction (*Juntti and Fernald, 2016*). We can thus propose that the putative neurosecretory cell type associated with the germ line that regulated gamete release in ancestral metazoans was also photosensitive. The active photopigments in these cells could have been cryptochromes, the most ancient metazoan photopigment family inherited from unicellular ancestors (*Cronin and Porter, 2014*; *Gehring, 2014*) or animal opsins, whose gene family is absent in sponges but expanded extensively in both cnidarians and bilaterians from a common ancestral GPCR gene (*Feuda et al., 2014*; *2012*; *Gehring, 2014*; *Liegertová et al., 2015*). Under this evolutionary scenario, the essential Opsin9 protein in *Clytia*, and similarly the gonad-expressed *Cladonema* (*Suga et al., 2008*) and *Tripedalia* (*Liegertová et al., 2015*) opsins, would have replaced the ancestral photopigment in the MIH-secreting cells during cnidarian evolution to provide optimised spawning responses to particular light wavelength and intensity. A parallel can be drawn with the evolution of vision in eumetazoans, in which deployment of animal opsins is thought to have allowed more rapid and precise photoresponses than the ancestral cryptochrome system (*Gehring, 2014*).

While the scenario above conforms to the evolutionary trend for specialised cell types with distributed functions in bilaterians to evolve from ancestral multifunctional cell types (*Arendt, 2008*; *Arendt et al., 2016*), an alternative hypothesis is that the *Clytia* gonad photosensitive–neurosecretory cells and deep brain photoreceptors in bilaterian species arose convergently during evolution. Specifically, a population of MIH-secreting cells in hydrozoan gonads, initially regulated by other environmental and/or neural inputs, may have secondarily acquired opsin expression to become directly photosensitive. More widely, cnidarians may have accumulated a variety of multifunctional cell types as specific populations of neural cells, muscle cells or nematocytes acquired photopigments during evolution (*Porter et al., 2012*). Cnidarians are characterised by the lack of a centralised nervous system, and correspondingly show localised regulation of many physiological processes and behaviours at the organ, tissue or even the cellular levels. In addition to gamete release, other local light-mediated responses include light-sensitive discharge of cnidocyte-associated sensory-motor neurons expressing hydra *HmOps2* (*Plachetzki et al., 2012*), diel cycles of swimming activity (*Martin, 2002*; *Mills, 1983*) controlled in *Polyorchis* by photoresponsive cells of the inner nerve ring (*Anderson and Mackie, 1977*), and tentacle retraction in corals (*Gorbunov and Falkowski, 2002*). *Clytia* Opsin9, the first cnidarian opsin to have a demonstrated function, is expressed in the scattered, MIH-secreting cells, which act autonomously in the gonad without any need for input from other parts of the medusa. It remains to be determined if they secrete the MIH peptides from localised cellular sites and are connected by synapses. The nature of these cells may thus be neurosecretory or neuroendocrine, rather than neuronal (*Hartenstein, 2006*), fitting with the idea that neuropeptide signalling between epithelial cells may have predated nervous system evolution (*Bosch et al., 2017*).

Whether Hydrozoa gonad ectoderm MIH-secreting cells and Bilateria deep brain photosensitive cells derived from an ancestral multifunctional photosensory-secretory cell type, or from non-photoresponsive neural cells remains an open question. Our findings support a scenario in which expansion of the opsin gene family within the hydrozoan clade was accompanied by the local deployment of individual opsins with specific spectral characteristics adjusted to regulate a variety of physiological behaviours in response to light, epitomised by *Clytia* Opsin9 and its regulation of spawning.

## Materials and methods

**Key resources table**

| Reagent type (species) or resource | Designation | Source or reference | Identifiers | Additional information |
|---|---|---|---|---|
| gene (*Clytia hemishaerica*) | *CheOpsin4* | this study | GenBank MF435920 | sequence retrieved from Clytia trancriptome data |

*Continued on next page*

*Continued*

| Reagent type (species) or resource | Designation | Source or reference | Identifiers | Additional information |
|---|---|---|---|---|
| gene (*C. hemisphaerica*) | *CheOpsin7* | this study | GenBank MF435921 | sequence retrieved from Clytia trancriptome data |
| gene (*C. hemisphaerica*) | *CheOpsin9* | this study | GenBank MF435922 | sequence retrieved from Clytia trancriptome data |
| genetic reagent (*C. hemisphaerica*) | *Opsin9* [n1–4] | this study | | Mosaic F0 polyp colony |
| antibody | anti-PRPamide | *Takeda et al., 2018* | | Affinity purified Rabbit polyclonal; against synthetic PRPamide |
| antibody | anti-acetylated alpha-tubulin (rat monoclonal) | Sigma Aldrich | 6-11B-1 | (1:500) |
| antibody | anti alpha-tubulin (rat monoclonal) | Thermo Fisher Scientific | YL1/2 | (1:1000) |
| antibody | anti- ɣ-Tubulin | Sigma Aldrich | GTU-88, | (1:200) |
| antibody | Rhodamine goat anti-rabbit, Cy5 goat anti-mouse-IgG, Cy5 goat anti-rabbit- secondaries | Jackson ImmunoResearch | | (1:1000) |
| other | rhodamine phalloidin | Molecular Probes | | (1:250) |
| other | Hoechst dye 33258 | Sigma Aldrich | | (1:5000) |
| guide RNA (sgRNA) | Opsin9 n1 | this study | | sequence given in *Figure 3—source data 1* |
| recombinant DNA reagent | pDR27457 (plasmid) | Addgene | 42250 | |
| Transcriptome data | *Clytia hemisphaerica* reference transcriptome | This study | GSE101072_Clytia_tophat_trinityGG_combined_v2 | Combined assembled transcriptome data from multiple samples. |
| Transcriptome data | *Clytia* gonad tissue samples | This study | GSM2698879 GEN-BR1 GSM2698880 GEN-BR2 GSM2698881 GEC-BR1 GSM2698882 GEC-BR2 GSM2698883 GrOo-BR1 GSM2698884 GrOo-BR2 GSM2698885 FGOo-BR1 GSM2698886 FGOo-BR2 | Illlumina HiSeq 50nt reads for 8 RNA samples (four tissues, two duplicates for each tissue ) |

## Animals

Sexually mature medusae from laboratory maintained *Clytia hemisphaerica* ('Z colonies') were fed regularly with *Artemia* nauplii and cultured under light-dark cycles to allow daily spawning. Red Sea Salt brand artificial seawater (ASW) was used for all culture and experiments.

## Monochromator assay

Manually dissected *Clytia* ovaries in small plastic petri dishes containing Millipore filtered sea water (MFSW) were maintained overnight in the dark and then stimulated with monochromatic light, provided by a monochromator (PolyChrome II, Till Photonics) installed above the samples, using the set-up described by *Gühmann et al. (2015)*, which delivers equivalent levels of irradiance between 400 and 600 nm (3.2E + 18 to 4.3E + 18 photons/s/m2). Monochromatic light excitation was carried out in a dark room. 10 s pulses of different wavelengths, between 340 to 660 nm were applied to separate groups of 3–6 gonads, which were then returned to darkness for one hour before monitoring of oocyte maturation seen as loss of the oocyte nucleus (Germinal Vesicle) in fully grown oocytes, followed by spawning. A wavelength was considered to induce maturation if at least one oocyte per gonad underwent maturation and spawning within 30 min of monochromatic light excitation. 10 s exposure times were chosen because initial trials showed that these gave sub-saturating responses at all wavelengths. Gonads that spawned prematurely due to manipulation stress were excluded from analysis.

## Identification of *Clytia* opsin genes

BLAST searches were performed on an assembled *Clytia hemisphaerica* mixed-stage transcriptome containing 86,606 contigs, using published cnidarian opsin sequences or *Clytia* opsin sequences as bait. The ORFs of selected *Clytia* opsins were cloned by PCR into pGEM-T easy vector for synthesis of in situ hybridisation probes.

## Gonad transcriptome analysis

Illlumina HiSeq 50nt reads were generated from mRNA isolated using RNAqueous micro kit (Ambion Life technologies, CA) from ectoderm, endoderm, growing oocytes and fully grown oocytes manually dissected from about 150 *Clytia* female gonads. The data have been deposited in NCBI's Gene Expression Omnibus and are accessible through GEO Series accession number GSE101072 (https://www.ncbi.nlm.nih.gov/geo/query/acc.cgi?acc=GSE101072) (*Quiroga Artigas et al., 2017*). Biological replicates for each sample consisted of pooled tissue fragments or oocytes dissected on different days, using the same conditions and jellyfish of the same age. The reads were mapped against the opsin sequences retrieved from a *Clytia* reference transcriptome using Bowtie2 (*Langmead and Salzberg, 2012*). The counts for each contig were normalised per total of reads of each sample and per sequence length. Opsins RNA read counts from each tissue were visualised as a colour coded heat map using ImageJ software.

## In situ hybridisation

For in situ hybridisation, isolated gonads were processed as previously (*Lapébie et al., 2014*) except that 4M Urea was used instead of 50% formamide in the hybridisation buffer as it significantly improves signal detection and sample preservation in *Clytia* medusa (*Sinigaglia et al., 2017*). Images were taken with an Olympus BX51 light microscope. For double fluorescent in situ hybridisation, female *Clytia* gonads were fixed overnight at 18°C in HEM buffer (0.1 M HEPES pH 6.9, 50 mM EGTA, 10 mM MgSO$_4$) containing 3.7% formaldehyde, washed five times in PBS containing 0.1% Tween20 (PBS-T), then dehydrated on ice using 50% methanol/PBS-T then 100% methanol. In situ hybridisation was performed using a DIG-labeled probe for Opsin9 and a fluorescein-labeled probe for PP4. A three hours incubation with a peroxidase-labeled anti-DIG antibody was followed by washes in MABT (100 mM maleic acid pH 7.5, 150 mM NaCl, 0.1% Triton X-100). For Opsin9 the fluorescence signal was developed using the TSA (Tyramide Signal Amplification) kit (TSA Plus Fluorescence Amplification kit, PerkinElmer, Waltham, MA) and Cy5 fluorophore (diluted 1/400 in TSA buffer: PBS/H$_2$O$_2$0.0015%) at room temperature for 30 min. After three washes in PBS-T, fluorescence was quenched with 0.01N HCl for 10 min at room temperature and washed again several times in PBS-T. Overnight incubation with a peroxidase-labelled anti-fluorescein antibody was followed by washes in MABT. The anti PP4 fluorescence signal was developed using TSA kit with Cy3 fluorophore. Controls with single probes were done to guarantee a correct fluorescence quenching and ensure that the two channels did not cross-over. Nuclei were stained using Hoechst dye 33258. Images were acquired using a Leica SP5 confocal microscope and maximum intensity projections of z-stacks prepared using ImageJ software (*Schneider et al., 2012*). Single in situ hybridisations were performed, and gave equivalent results, three times and double in situ hybridisations twice.

## Immunofluorescence

For co-staining of neuropeptides and tyrosinated tubulin, dissected *Clytia* gonads were fixed overnight at 18°C in HEM buffer containing 3.7% formaldehyde, then washed five times in PBS containing 0.1% Tween20 (PBS-T). Treatment on ice with 50% methanol/PBS-T then 100% methanol plus storage in methanol at −20°C improved visualisation of microtubules in the MIH-producing cells. Samples were rehydrated, washed several times in PBS-0.02% Triton X-100, then one time in PBS-0.2% Triton X-100 for 20 min, and again several times in PBS-0.02% Triton X-100. They were then blocked in PBS with 3% BSA overnight at 4°C. The day after they were incubated in anti-PRPamide antibody and anti-Tyr tubulin (YL1/2, Thermo Fisher Scientific) in PBS/BSA at room temperature for 2 hr. After washes, the specimens were incubated with secondary antibodies (Rhodamine goat anti-rabbit and Cy5 donkey anti-rat-IgG; Jackson ImmunoResearch, West Grove, PA) overnight in PBS at 4°C, and nuclei stained using Hoechst dye 33258 for 20 min.

For co-staining of neuropeptides with cortical actin, cilia (acetylated α-tubulin) or cilary basal bodies (ɣ-Tubulin), dissected *Clytia* gonads were fixed for 2–3 hr at room temperature in HEM buffer containing 80 mM maltose, 0.2% Triton X-100% and 4% paraformaldehyde, then washed five times in PBS containing 0.1% Tween20 (PBS-T). Samples were further washed in PBS-0.02% Triton X-100, then one time in PBS-0.2% Triton X-100 for 20 min, and again several times in PBS-0.02% Triton X-100. They were then blocked in PBS with 3% BSA overnight at 4°C. The day after they were incubated in anti-PRPamide antibody and combinations of anti- ɣ-Tubulin (GTU-88, Sigma Aldrich), anti-acetylated α-tubulin (6-11B-1, Sigma Aldrich), in PBS/BSA at room temperature for 2 hr. After washes, the specimens were incubated with appropriate combinations of secondary antibodies (Rhodamine or Cy5 goat anti-rabbit; fluorescein or Cy5 goat anti-mouse-IgG) and Rhodamine-Phalloidin overnight in PBS at 4°C, and nuclei stained using Hoechst dye 33258 for 20 min. Images were acquired using a Leica SP8 confocal microscope and maximum intensity projections of z-stacks prepared using ImageJ software.

For MIH fluorescence quantification, 5–6 independent gonads for each of the two conditions (light-adapted and dark-adapted after light stimulation) and *Clytia* strains (WT and *Opsin9*[n1–4]) were fixed as mentioned above and co-stained for MIH and Tyr-tubulin. All the fixations were done in parallel. Confocal images were acquired using the same scanning parameters (i.e. magnification, laser intensity and gain). In all cases, 10 confocal Z planes were summed over 4 µm depth at the gonad surface using ImageJ software. With ImageJ, we separated the two channels (MIH and Tyr-tubulin) and selected the contour of MIH-positive cells using the Tyr-tubulin staining as guidance. Using the 'Integrated Density' option, we recovered the 'RawIntDen' values of the MIH-stained channel, which refer to the sum of the pixel intensity values in the selected region of interest. These values divided by 1000 correspond to the RFU (Relative Fluorescence Units) in *Figure 4*.

## Generation of CRISPR-Cas9 mutant *Clytia* colonies

The template for *Opsin9* n1 small guide RNA (*Opsin9* n1 sgRNA; sequence in *Figure 3—source data 1*) was assembled by cloning annealed oligonucleotides corresponding 20 bp Opsin9 target sequence into pDR274 (*Hwang et al., 2013*) (42250, Addgene), which contains tracrRNA sequence next to a BsaI oligonucleotide insertion site. The sgRNA was then synthesised from the linearised plasmid using Megashortscript T7 kit (Thermo Fisher Scientific) and purified with ProbeQuant G-50 column (GE healthcare) and ethanol precipitation. The sgRNA was dissolved in distilled water at 80 µM and kept at −80°C until use. Purified Cas9 protein dissolved in Cas9 buffer (10 mM Hepes, 150 mM KCl) was kindly provided by J-P Concordet (MNHN Paris) and diluted to 5 µM. The sgRNA was added to Cas9 protein in excess (2:1) prior to injection and incubated for 10 min at room temperature. The final Cas9 concentration was adjusted to 4.5 µM and the sgRNA to 9 µM. The mixture was centrifuged at 14,000 rpm for 10 min at room temperature. 2–3% of egg volume was injected into unfertilised eggs within 1 hr after spawning, prior to fertilisation.

Injected embryos were cultured for 3 days in MFSW at 18°C. Metamorphosis of planula larvae into polyps was induced about 72 hr after fertilisation by placing larvae (20–80/slide) on double-width glass slides (75 × 50 mm) in drops of 3–4 ml MFSW containing 1 µg/ml synthetic metamorphosis peptide (GNPPGLW-amide; Genscript), followed by overnight incubation. Slides with fixed primary polyps were transferred to small aquariums kept at 24°C, to favour the establishment of female colonies (*Carré and Carré, 2000*). Primary polyps and young polyp colonies were fed twice a day with smashed *Artemia* nauplii until they were big enough to be fed with swimming nauplii. Following colony vegetative expansion, a single well-growing colony on each slide was maintained as a founder. After several weeks of growth, polyp colonies were genotyped to assess mutation efficiency and mosaicism, and medusae were collected from the most strongly mutant colony (colony number four obtained using the n1 guide RNA, designated *Opsin9*[n1–4]) for further experimentation.

## Mutant genotyping

Genomic DNA from *Clytia* polyps and jellyfish gonads was purified using DNeasy blood/tissue extraction kit (Qiagen). The *opsin9* target site was amplified by PCR using Phusion DNA polymerase (New England Biolabs). Primers used for genotyping are listed in *Figure 3—source data 1* . PCR products were sequenced and mutation efficiency was assessed using TIDE analyses (*Figure 3—*

*figure supplement 1*), which estimates the mutation composition from a heterogeneous PCR product in comparison to a wild type sequence (*Brinkman et al., 2014*).

We scanned *Clytia* genome for possible off-targets of *Opsin9* sgRNA at http://crispor.tefor.net. From 2 possible off-targets where Cas9 could cut, none was found in coding sequences nor were they right next to a PAM sequence.

## Gonad spawning assays

Sexually mature wild type and *Opsin9*[n1–4] mutant medusae of the same age and adapted to the same day-night cycle were collected for gonad dissection. Individual gonads were transferred to 100 μl MFSW in 96-well plastic plates. Plates were covered with aluminium foil overnight and brought back to white light the following day. For the rescue experiment with synthetic MIH, wild type and *opsin9*[n1–4] mutant gonads adapted to light conditions were dissected and transferred to 96-well plastic plates and acclimatised for two hours. An equal concentration (100 μl) of double concentrated ($2 \times 10^{-7}$M) synthetic WPRPamide (synthetic MIH; Genscript) in MFSW was added in each well to give a final concentration of $10^{-7}$M. Oocyte maturation was scored after one hour. Spawning followed in all cases where oocyte maturation was triggered. Gonads that spawned prematurely due to manipulation stress were excluded from analysis. Gonad pictures in *Figure 3* were taken with an Olympus BX51 microscope.

## Graphs and statistics

Graphs and statistics for the monochromator assay were prepared using BoxPlotR (*Spitzer et al., 2014*). Fisher's exact tests and Mann-Whitney U tests were performed at http://www.socscistatistics.com. Fisher's exact tests were chosen for analysing the spawning results of *Figure 3D,E* based on 2 × 2 contingency tables. Nonparametric Mann-Whitney U tests were chosen for the MIH fluorescence quantification comparisons (*Figure 4*) since the results did not follow a normal distribution according to the Shapiro-Wilk test and significance between sample distributions could be appropriately assessed by data transformation into ranks.

## Opsin molecular phylogeny

To assess the relationship of the *Clytia* opsin amino acid sequences to known opsins, we added them to two recent datasets (*Vöcking et al., 2017*; *Feuda et al., 2014*). All incomplete sequences were removed from the untrimmed *Vöcking et al. (2017)* dataset before adding the *Clytia* opsin sequences using profile alignment in MUSCLE (*Edgar, 2004*). An Opsin sequence found in our unpublished *Melicertum octocostatum* transcriptome (Leptothecata, Hydrozoa) was added to this dataset using profile alignment. These untrimmed alignments were used in phylogenetic analyses. The same procedure was used for adding the *Clytia* opsin sequences to the *Feuda et al. (2014)* trimmed dataset. Positions containing only *Clytia* opsin sequences were removed. For more detailed comparison between medusozoan sequences, we generated an alignment including all cubozoan and hydrozoan available opsin protein sequences in GenBank (March 2017) and the 10 *Clytia* opsin sequences. Cd-hit (*Fu et al., 2012*) was run with 99% identity to eliminate sequence duplicates, obtaining a final dataset of 56 Cubozoa and Hydrozoa opsin protein sequences. The alignment was performed using MUSCLE (*Edgar, 2004*) and further adjusted manually in Seaview v4.2.12 (*Galtier et al., 1996*) to remove the ambiguously aligned N- and C-terminal regions, as well as positions including only one residue. Alignments used for phylogenetic analyses are available as *Figure 6—source data 1*.

Maximum likelihood (ML) analyses were performed using RaxML 8.2.9 (*Stamatakis, 2014*). The GTR+Γ and LG+Γ models of protein evolution were used following *Feuda et al. (2014)* and *Vöcking et al. (2017)* with parsimony trees as starting trees. ML branch support was estimated using non-parametric bootstrapping (500 replicates). Bayesian analyses (MB) were carried out using MrBayes 3.2.6 (*Ronquist et al., 2012*), with the LG+Γ model, performing two independent runs of four chains for 1 million generations sampled every 100 generations. MB analyses were considered to have converged when average standard deviation of split frequencies dropped below 0.05. Consensus trees and posterior probabilities were calculated using the 200.000 last generations. The resulting trees were visualised with FigTree ( http://tree.bio.ed.ac.uk/software/figtree/).

Approximated Unbiased (AU) phylogenetic tests (*Shimodaira, 2002*) were performed as described previously (*Leclère and Rentzsch, 2012*). Values presented in the Results section were obtained comparing, among others, the ML tree with *Clytia* Opsin9, Opsin10 and either *Cladonema* or *Tripedalia* gonad-expressed Opsins constrained as monophyletic using RaxML. Exclusion of the non-gonad-expressed Opsin10 from these monophyletic constraints led to much lower p values (not shown).

## Acknowledgements

We gratefully acknowledge the technical assistance of J Uveira and S Chevalier, as well as L Gissat and Sophie Collet for animal maintenance. We thank M Khamla for the graphics in *Figure 1A*, J-P Concordet (MNHN Paris) for generously providing Cas9 protein, M Gühmann for monochromator training, C Rouvière for image quantification methods and all our laboratory and NEPTUNE colleagues for stimulating and critical discussions. Funding was provided by the Marie Curie ITN NEPTUNE, French ANR grant OOCAMP, EMBRC-Fr infrastructure development funding and core CNRS funding to the LBDV. Microscopy was performed at the PIV imaging platform.

## Additional information

### Funding

| Funder | Grant reference number | Author |
|---|---|---|
| European Commission | FP7-PEOPLE-2012-ITN 317172 (NEPTUNE) | Gáspár Jékely<br>Evelyn Houliston |
| Centre National de la Recherche Scientifique | | Evelyn Houliston |
| Agence Nationale de la Recherche | ANR- 13-BSV2-0008-01 ("OOCAMP") | Evelyn Houliston |

The funders had no role in study design, data collection and interpretation, or the decision to submit the work for publication.

### Author contributions

Gonzalo Quiroga Artigas, Conceptualization, Investigation, Writing—original draft, Writing—review and editing; Pascal Lapébie, Conceptualization, Data curation, Software, Investigation, Writing—review and editing; Lucas Leclère, Conceptualization, Formal analysis, Writing—review and editing; Noriyo Takeda, Ryusaku Deguchi, Resources, Writing—review and editing; Gáspár Jékely, Supervision, Writing—review and editing; Tsuyoshi Momose, Conceptualization, Supervision, Investigation, Writing—review and editing; Evelyn Houliston, Conceptualization, Supervision, Writing—original draft, Writing—review and editing

### Author ORCIDs

Lucas Leclère (iD) http://orcid.org/0000-0002-7440-0467
Ryusaku Deguchi (iD) http://orcid.org/0000-0003-4571-9329
Gáspár Jékely (iD) http://orcid.org/0000-0001-8496-9836
Tsuyoshi Momose (iD) https://orcid.org/0000-0002-3806-3408
Evelyn Houliston (iD) http://orcid.org/0000-0001-9264-2585

### Decision letter and Author response

Decision letter https://doi.org/10.7554/eLife.29555.024
Author response https://doi.org/10.7554/eLife.29555.025

# Additional files

## Supplementary files
• Transparent reporting form

DOI: https://doi.org/10.7554/eLife.29555.020

## Major datasets
The following dataset was generated:

| Author(s) | Year | Dataset title | Dataset URL | Database, license, and accessibility information |
|---|---|---|---|---|
| Quiroga Artigas G, Lapébie P, Leclère L, Houliston E | 2017 | RNA-Seq of different Clytia female gonad tissues and oocytes at different growth stage | https://www.ncbi.nlm. nih.gov/geo/query/acc. cgi?acc=GSE101072 | Publicly available at the NCBI Gene Expression Omnibus (accession no: GSE10 1072) |

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
