## [Decision Letter]

Thank you for submitting your article "CRISPR/Cas9 mutation of a gonad-expressed opsin prevents jellyfish light-induced spawning" for consideration by *eLife*. Your article has been reviewed by two peer reviewers, and the evaluation has been overseen by Alejandro Sánchez Alvarado as the Reviewing Editor and K VijayRaghavan as the Senior Editor. The following individuals involved in review of your submission have agreed to reveal their identity: Laurinda A. Jaffe (Reviewer #2); Steve Haddock (Reviewer #3).

The reviewers have discussed the reviews with one another and the Reviewing Editor has drafted this decision to help you prepare a revised submission.

Summary:

This is an outstanding study that provides exciting new information about a long-standing question in the field of developmental biology: how external signals cause eggs and sperm to be released from marine animals, allowing fertilization and development. Dark/light transitions are known to trigger spawning in a variety of species, but this work is the first to identify the cell and opsin that mediate secretion of a hormone that acts on the oocyte to cause its release from the ovary. This signaling also stimulates the oocyte meiotic cell cycle, in preparation for fertilization. The experiments supporting these conclusions are very clear and are beautifully documented. The reviewers were unanimously enthusiastic about quality of the work, particularly the clear and elegant experiments reported.

Minor points:

Although this paper is generally very well written with excellent figures, we have some minor suggestions for improving the presentation listed below.

Consider a shorter title, without "CRISPER/Cas 9 mutation", such as "A gonad-expressed opsin mediates light-induced spawning in jellyfish". The original title is also fine if the authors want to emphasize the novel way that this method was used. If the method is to be included in the title we strongly suggest it be changed to: "CRISPER/Cas 9 mutation of a gonad-expressed opsin prevents light-induced spawning in jellyfish".

The section on opsin phylogeny might be better placed at the end of the Results, with a separate subheading. In the present manuscript, the description of Figure 2 is broken up by two paragraphs about Figure 3. Also, the sentence about the absence of detectable opsin 9 expression in non-gonad tissues (subsection “Opsin9 gene knockout prevents oocyte maturation and spawning”, last paragraph) would be better mentioned in connection with Figure 2.

Figure 3 should be explained for a more general audience. Mention the functions of the various types of opsins (rhabdomeric, retinal, ciliary). Add a phrase to define "bilateria". Add a reference for the statement that "opsin sequence evolution is rapid".

Opsin 9^n1_4^ should be defined in the Results.

The finding that treatment of Opsin 9 KO ovaries with synthetic MIH peptides (subsection “Opsin9 is required for light-induced MIH secretion from gonad ectoderm”) would be better mentioned in the preceding section. At the end of the second paragraph of the section on the prevention of oocyte maturation and spawning, add something like the following: "However, these gonads did release oocytes in response to synthetic MIH peptide (Figure 4). These oocytes resumed meiosis normally and could be fertilized […]"

Consider adding a photo (or a sentence) indicating that the oocytes that remained in the ovary in the Opsin 9 KO failed to resume meiosis in the ovary.

For the section on the opsin requirement for the light-induced MIH secretion (subsection “Opsin9 is required for light-induced MIH secretion from gonad ectoderm”), the second line of evidence seems to be the important one.

The description of "neuronal morphology" is based on the observation of "neural processes […] that did not appear to form a connected network". Because further studies would be needed to determine whether these cells are neurons (as defined as cells that are connected to the rest of the nerve net), consider replacing the term neuron, or if it is used, explain that this is a tentative terminology. However, this is not a major issue. The findings are equally interesting whether or not these cells are "neurons".

The statement that the MIH-secreting cells "appear to respond individually to light with no obvious coordination of MIH release between them" should be more completely explained.

References 24 and 32 are the same (Fischer et al., 2013). Reference 56 (Sinigaglia et al., 2017) needs the journal information.

The color-coding in Figure 4 could be improved.

Consider presenting some higher magnification images in Figure 6.

Throughout: don't use one parenthesis in figure legends especially when there are other parentheses included. (Use (1) not 1)).

Maybe cite the recent review by Siebert on hydrozoan reproduction.

Labels for Trees in Figure 3—figure supplement 1 and 2 are unintelligible.

---

## [Author Response]

Minor points:Although this paper is generally very well written with excellent figures, we have some minor suggestions for improving the presentation listed below.Consider a shorter title, without "CRISPER/Cas 9 mutation", such as "A gonad-expressed opsin mediates light-induced spawning in jellyfish". The original title is also fine if the authors want to emphasize the novel way that this method was used. If the method is to be included in the title we strongly suggest it be changed to: "CRISPER/Cas 9 mutation of a gonad-expressed opsin prevents light-induced spawning in jellyfish".

We agree with this suggestion and have changed the title to: “A gonad-expressed opsin mediates light-induced spawning in the jellyfish *Clytia*”.

The section on opsin phylogeny might be better placed at the end of the Results, with a separate subheading. In the present manuscript, the description of Figure 2 is broken up by two paragraphs about Figure 3.

We followed this advice. The Opsin phylogeny section is now the subsection “Evolution of the *Clytia* Opsin9 gene”, and Figure 3 has been correspondingly repositioned and renamed as Figure 6.

Also, the sentence about the absence of detectable opsin 9 expression in non-gonad tissues (subsection “Opsin9 gene knockout prevents oocyte maturation and spawning”, last paragraph) would be better mentioned in connection with Figure 2.

We have added a sentence: “No Opsin9 expression was detected anywhere else in the medusa”.

Figure 3 should be explained for a more general audience. Mention the functions of the various types of opsins (rhabdomeric, retinal, ciliary).

We have extensively revised the section on opsin evolution (subsection “Evolution of the *Clytia* Opsin9 gene”) to take into account recent advances in the understanding of opsin evolution. Furthermore we re-analysed the phylogenetic relationships between the *Clytia* opsins and other animal opsins using the recently published sequence alignment of Vöcking et al. (2017), rather than the less comprehensive Feuda et al. (2014) dataset used in our original manuscript. Figure 6 and Figure 6—figure supplement 1 and 2 have been updated to show this new analysis, which did not affect the conclusions concerning *Clytia* Opsin9 evolution.

Add a phrase to define "bilateria".

Now defined at the end of the Introduction as “an animal clade comprising all the protostomes and deuterostomes”

Add a reference for the statement that "opsin sequence evolution is rapid".

In the revised version this statement has been eliminated; the whole opsin phylogeny section is now much more detailed and fully referenced.

Opsin 9^n1_4^ should be defined in the Results.

We have now explicitly flagged this designation in the Results (subsection “Opsin9 gene knockout prevents oocyte maturation and spawning”, first paragraph) and provided a full explanation in the Materials and methods (subsection “Opsin molecular phylogeny”).

The finding that treatment of Opsin 9 KO ovaries with synthetic MIH peptides (subsection “Opsin9 is required for light-induced MIH secretion from gonad ectoderm”) would be better mentioned in the preceding section. At the end of the second paragraph of the section on the prevention of oocyte maturation and spawning, add something like the following: "However, these gonads did release oocytes in response to synthetic MIH peptide (Figure 4). These oocytes resumed meiosis normally and could be fertilized […]"

This suggestion has been followed (subsection “Opsin9 gene knockout prevents oocyte maturation and spawning”, third paragraph).

Consider adding a photo (or a sentence) indicating that the oocytes that remained in the ovary in the Opsin 9 KO failed to resume meiosis in the ovary.

This information has been added in the subsection “Opsin9 gene knockout prevents oocyte maturation and spawning”.

For the section on the opsin requirement for the light-induced MIH secretion (subsection “Opsin9 is required for light-induced MIH secretion from gonad ectoderm”), the second line of evidence seems to be the important one.

We agree with this and so have moved information from the first part of this paragraph, concerning the ability of exogenous peptides to ‘rescue’ Opsin9 mutant spawning to an earlier paragraph, as noted also above.

The description of "neuronal morphology" is based on the observation of "neural processes […] that did not appear to form a connected network". Because further studies would be needed to determine whether these cells are neurons (as defined as cells that are connected to the rest of the nerve net), consider replacing the term neuron, or if it is used, explain that this is a tentative terminology. However, this is not a major issue. The findings are equally interesting whether or not these cells are "neurons".

We agree that the issue of whether the MIH-secreting cells are connected remains to be resolved and have added sentences to this effect in the Results (subsection “Neural type morphology of *Clytia* gonad MIH-secreting cells”), as well as in the Discussion (last paragraph). We have also taken care not to refer to these cells as ‘neural’ or ‘neuronal’ anywhere in the manuscript, using instead using terms such as "cells with neural-type morphology" or "sensory-neurosecretory cells".

The statement that the MIH-secreting cells "appear to respond individually to light with no obvious coordination of MIH release between them" should be more completely explained.

To avoid possible over interpretation of this non-quantified observation we have removed this sentence and replaced with a short discussion emphasising that connections between MIH/Opsin9 cells remains to be demonstrated, and that the nature of these cells may be more neurosecretory / neuroendocrine cells than neuronal (as distinguished for instance by Hartenstein, 2006).

References 24 and 32 are the same (Fischer et al., 2013). Reference 56 (Sinigaglia et al., 2017) needs the journal information.

The reference list has been reformatted and thoroughly checked.

The color-coding in Figure 4 could be improved.

We have modified panel A to brighter and clearer colours and more obvious matching of the 7tm protein cartoon and the CRISPR target sequence.

Consider presenting some higher magnification images in Figure 6.

We have extensively reconstructed this figure (now Figure 5) to include higher magnification images for most of the staining combinations involved. We have also clarified the accompanying text, especially with respect to the unresolved issue of whether the MIH/Opsin9 cells have an exposed apical surface and apical cilia.

Throughout: don't use one parenthesis in figure legends especially when there are other parentheses included. (Use (1) not 1)).

Parenthesis use in the legends has been clarified.

Maybe cite the recent review by Siebert on hydrozoan reproduction.

This review is now cited in the Introduction.

Labels for Trees in Figure 3—figure supplement 1 and 2 are unintelligible.

We have completely replaced all the Opsin phylogeny figures (Figure 6 and Figure 6—figure supplement 1 and Figure 6—figure supplement 2) with new, more comprehensive, analyses and clearer labels.